# Risk Factors for Malignant Transformation in Inverted Sinonasal Papilloma: A Systematic Review and Meta-Analysis

**DOI:** 10.3390/cancers17111798

**Published:** 2025-05-28

**Authors:** Andrea Ambrosini-Spaltro, Giulia Querzoli, Anna Caterina Leucci, Angela Camagni, Paolo Farneti, Elisa D’Angelo, Elisa Donini, Alicia Tosoni, Ernesto Pasquini, Paolo Galli, Maria P. Foschini

**Affiliations:** 1Pathology Unit, Morgagni-Pierantoni Hospital, AUSL Romagna, 47121 Forlì, Italy; 2Pathology Unit, IRCCS Azienda Ospedaliero Universitaria di Bologna, 40138 Bologna, Italy; giulia.querzoli2@unibo.it; 3Department of Medical and Surgical Sciences (DIMEC), University of Bologna, 40138 Bologna, Italy; 4Department of Biomedical and Neuromotor Sciences (DIBINEM), Alma Mater Studiorum, University of Bologna, 40139 Bologna, Italy; annacaterina.leucci3@unibo.it; 5Sinonasal Cancer Registry of Emilia Romagna, Occupational Safety and Prevention Unit, Public Health Department, Bologna Local Health Authority, 40121 Bologna, Italy; angela.camagni@ausl.bologna.it (A.C.); paolo.galli@ausl.bologna.it (P.G.); 6ENT Unit, Bellaria Hospital, AUSL Bologna, 40139 Bologna, Italy; paolo.farneti@ausl.bologna.it (P.F.); ernesto.pasquini@ausl.bologna.it (E.P.); 7Radiotherapy—AUSL di Bologna-Ospedale Bellaria, 40139 Bologna, Italy; elisa.dangelo@ausl.bologna.it (E.D.); e.donini@ausl.bologna.it (E.D.); 8Nervous System Medical Oncology Department, IRCCS Istituto delle Scienze Neurologiche di Bologna, 40139 Bologna, Italy; a.tosoni@isnb.it; 9Unit of Anatomic Pathology, Bellaria Hospital, Department of Biomedical and Neuromotor Sciences (DIBINEM), Alma Mater Studiorum, University of Bologna, 40139 Bologna, Italy; mariapia.foschini@unibo.it

**Keywords:** inverted papilloma, carcinoma, sinonasal, HPV, smoking, alcohol, meta-analysis

## Abstract

Inverted sinonasal papilloma (IP) is a benign proliferation that can recur and undergo malignant transformation. We conducted a systematic review and meta-analysis to identify the risk factors associated with malignant transformation. After collecting 1875 articles, we identified a population of 1271 inverted papillomas and performed three different meta-analyses: the significant risk factors for malignant transformation were smoking (*p* = 0.002) and HPV (*p* < 0.001), whereas alcohol consumption was not significant (*p* = 0.95). Potential strategies include encouraging individuals with IPs to quit smoking and to receive the HPV vaccine.

## 1. Introduction

Sinonasal papilloma is a benign epithelial neoplasm of the sinonasal tract and is divided into three forms: inverted, oncocytic, and exophytic [1]. Inverted sinonasal papilloma (IP) is the most frequent type of papilloma, but it is still a rare disease, with an incidence between 0.2 and 1.5 cases per 100,000 persons per year [2].

IP may recur with a recurrence rate of 13.72% [3]. Malignant transformation may also occur, with carcinoma arising in a pre-existing inverted papilloma (C-IP), usually squamous cell carcinoma. The incidence of malignancy varies across different studies and is usually 7.6% [3]. The criteria to diagnose malignant transformation of IP are not clearly defined, nor are those for dysplasia in IP; however, carcinoma arising in an IP is usually easily diagnosed on histology. The most frequent histotypes arising in IP are squamous cell carcinoma and mucoepidermoid carcinoma [4] (Figure 1).

The risk factors favoring IP incidence and recurrence are not very clear. In IP incidence, outdoor and industrial occupations were associated with IP and may be potential risk factors [5]. Tobacco smoking and alcohol consumption did not seem to be significant risk factors for IP incidence [5]. However, tobacco smoking has been reported as a risk factor for IP recurrence [6].

Risk factors for IP malignant transformation have rarely and not extensively been examined in the literature. Three meta-analyses examined the role of human papillomavirus (HPV) in IP malignant transformation, indicating that HPV is an important risk factor in IP malignant transformation [7,8,9]. To the best of our knowledge, other factors have not been examined in systematic reviews or meta-analyses.

The purpose of this study is to perform a systematic review and meta-analysis to identify the risk factors associated with IP malignant transformation.

## 2. Materials and Methods

### 2.1. Guidelines and PICO

The present study followed the guidelines of the Preferred Reporting Items for Systematic Reviews and Meta-Analyses (PRISMA) 2020 statement [10]. The PRISMA checklist with the requested information is available in the Appendix A. We described the PICO elements (population, intervention/index, comparison, and outcome) as follows:Participants: patients with IP malignant transformation;Intervention/Index: risk factor exposure;Comparison: IP without malignant transformation;Outcome: number of malignant transformations in exposed vs. non-exposed individuals.

### 2.2. Protocol Registration

Before beginning the search, the present protocol was recorded in Prospero, a known portal for systematic reviews and meta-analyses, under the registration number CRD42024492228, available from https://www.crd.york.ac.uk/PROSPERO/view/CRD42024492228 (last accessed and verified on 6 April 2025).

### 2.3. Search Strategy

On 8 January 2024, studies were searched using two databases, PubMed and Embase. The search strategy was based on the following question: what are the risk factors favoring IP malignancy? To answer this question, the two main search criteria were combined:sinonasal inverted papilloma;malignant transformation.

We used a combination of keywords and controlled vocabulary. The controlled vocabulary is composed of MeSH terms in PubMed and Emtree terms in Embase.

The following text was used for the two databases:1.PubMed

(paranasal sinuses[MeSH Terms] OR sinonasal[All Fields]) AND (inverted papilloma[MeSH Terms] OR (inverted[All Fields] AND papilloma[All Fields])) AND (malignancy OR carcinoma)

2.Embase

inverted AND (‘papilloma’/exp OR papilloma) AND (paranasal OR sinonasal) AND (malignan* OR carcinom*)

### 2.4. Selection of Articles

All citations were obtained from the two different databases (PubMed and Embase) and their results were imported into the online portal Rayyan, https://www.rayyan.ai. Both databases were accessed and searched on 8 January 2024.

Duplicates were suggested by the Rayyan portal and were all controlled by one of the authors (A.A.-S.). One author (A.A.-S.) screened articles (some by abstract and title alone, some by entire full text, and eventually by Appendix A); a considerable number of papers was discussed together by the authors (A.A.-S., G.Q.). In cases of disagreement, a consensus was reached for each case.

### 2.5. Eligibility Criteria

Articles were selected based on the following inclusion criteria:

Observational and interventional studies, cohort studies, in English, with any publication date, from any country.Patients above 18 years of age presenting with a histologically proven diagnosis of sinonasal inverted papilloma and malignant transformation, either synchronous (present at the time of initial diagnosis) or metachronous (during follow-up).Documented exposure to risk factors with at least one of the following:
SmokingAlcoholInfectious agents (i.e., viruses)Professional exposure (i.e., exposure to chemicals or substances specifically related to a determined occupational activity, such as dusts or solvents).Comparison of the incidence of malignant transformation between exposed and non-exposed individuals. The comparison may be documented either by the number of individuals affected by malignancy compared to those who did not have malignancy or by odds ratio. Odds ratio values had to be clearly expressed in the main text, tables, figures, or Appendix A.

Articles that met the following exclusion criteria were excluded:Studies reporting case series with fewer than 5 participants.Only malignant transformed cases with no comparison with non-transformed (benign) cases.No adequate comparisons with non-exposed individuals.Studies with unclear diagnostic methods.Selected populations (i.e., only geriatric cases or only cases from a specific topography)Studies including non-consecutive cases, but only selected cases (possible selection biases).Foreign language (not English).Discussion papers (i.e., reviews, both narrative and systematic), with no new cases described.Cases including only carcinoma in situ with no invasive component in the evaluation of malignant transformation.

### 2.6. Risk of Bias Assessment

The Newcastle–Ottawa Scale (NOS) for non-randomized studies in meta-analyses was used to assess the risk of bias [11]. The Newcastle–Ottawa Scale (NOS) was developed for non-randomized studies, specifically designed for case-control studies, and is composed of three domains: (1) selection of groups, (2) comparability of groups, and (3) exposure. A “star system” evaluates each of the three domains as follows.

Selection (four items with a maximum of four stars).Is the case definition adequate?Representativeness of the casesSelection of controlsDefinition of controlsComparability (one item with a maximum of two stars)Comparability of cases and controls on the basis of the design or analysisExposure (three items with a maximum of three stars)Ascertainment of exposureSame method of ascertainment for cases and controlsNon-response rate

The maximum refers to the entire scoring system, and not specifically to the cases examined in our study. Two authors (A.A.-S. and G.Q.) independently evaluated the selected studies. In articles with disagreement, a consensus was reached for each case.

### 2.7. Statistical Analysis

The number of affected individuals in exposed versus non-exposed groups, or odds ratio values, was compared for each specific risk factor examined. Meta-analyses and forest plots were obtained using Review Manager (RevMan) software version 5.4 [12]. Meta-analyses were conducted using both the fixed-effects model and the random-effects model, in all comparisons. The heterogeneity of the results was assessed using the *I*^2^ statistic output, which was directly calculated by the RevMan software. Heterogeneity was classified as low, moderate, or high, with *I*^2^ values of 25%, 50%, and 75%, respectively [13]. Statistical significance (*p*) was set at 0.05, using a two-tailed hypothesis.

### 2.8. Quality of Evidence

To assess the overall quality of evidence, we used the Grading of Recommendations, Assessment, Development, and Evaluation (GRADE) approach [14]. GRADE uses a structured approach to assign ratings of high, moderate, low, or very low certainty to evidence. Since we considered all observational studies, the confidence started low. The evidence was then upgraded or downgraded separately for each outcome. The following factors may downgrade the level of evidence: (1) risk of bias, (2) inconsistency (or heterogeneity), (3) indirectness, (4) imprecision, and (5) publication bias. The following factors may upgrade the level of evidence: (1) large magnitude of effect, (2) dose–response gradient, and (3) direction of plausible bias. Each of the main factors may rate up or down the overall quality by one to two levels. The GRADE approach used in this observational study is illustrated in Figure 2.

## 3. Results

### 3.1. Article Selection

In total, 1875 results were obtained. The PubMed search retrieved 942 records. The Embase search retrieved 933 records. After removing 525 duplicates, 1350 citations were obtained. Of these, 1124 were excluded based only on title and abstract. In total, 226 citations were searched for full text and 195 full-text articles were examined.

After examining full texts, 3 studies were excluded because they examined only selected populations. Tong [15] did not consider consecutive cases, Eavey [16] examined only the pediatric population, and Elner [17] considered only cases invading the orbit. The study conducted by Valibeigi [18] was excluded because it considered both exophytic and inverted papillomas. All selected congress abstracts were excluded since they did not provide sufficient methodological details [19,20,21,22]. A total of 26 articles was selected [23,24,25,26,27,28,29,30,31,32,33,34,35,36,37,38,39,40,41,42,43,44,45,46,47,48]. Detailed information on article selection is summarized in the PRISMA flowchart (Figure 3).

### 3.2. Risk of Bias Assessment

To assess the risk of bias, the NOS tool analyzed all 26 selected studies, with a total score ranging from 7 to 8. A total of 26 articles underwent meta-analyses for at least one risk factor. The detailed results and risk of bias assessment are summarized in Table 1.

### 3.3. Population Examined

Among the 26 selected articles, the number of cases examined ranged from 14 to 162. Across all studies, a total population of 1271 inverted sinonasal papillomas was examined, among which 230 had carcinomas and 1041 did not. The global carcinoma incidence was 230/1271 (18.1%).

The clinicopathological features of the 26 selected articles are summarized in Table 2.

After searching for all risk factors listed in paragraph 2.5 (smoking, alcohol, infectious agents, professional exposure) in all articles, we identified the following risk factors: smoking, alcohol, and HPV. Only one study examined the role of EBV [29], and a comparison with a meta-analysis for EBV could not be performed. No other risk factor with sufficient data was found in the articles examined.

### 3.4. Meta-Analysis Results

Three different meta-analyses were performed for the following risk factors: smoking, alcohol, and HPV. Using the fixed-effects model, significant values were obtained for smoking (*p* = 0.002) and HPV (*p* < 0.001); alcohol did not reach statistical significance (*p* = 0.95). Using the random-effects model, only HPV reached statistical significance (*p* < 0.001), while both smoking (*p* = 0.22) and alcohol (*p* = 0.99) were not significant. Figure 4, Figure 5 and Figure 6 highlight forest plots and statistical data for HPV, smoking and alcohol, respectively, using both fixed-effects (insets A) and random-effects models (insets B).

The *I*^2^ values for HPV and alcohol (*I*^2^ = 0% in both) indicated very low heterogeneity, meaning that the results of the studies included in the meta-analysis for HPV and alcohol are highly consistent, with no significant differences between them. This suggests that any variations in the results are likely due to chance rather than differences in methodology, study populations, or other factors. For HPV, this finding reinforces its role as a significant risk factor for malignant transformation. Conversely, for alcohol, it confirms that it does not play a substantial role as a risk factor in this context, even if the small number of studies and patients examined may limit its evaluation.

Regarding smoking (*I*^2^ = 47%), the heterogeneity was classified as low to moderate (*I*^2^ is below 50%), indicating some variability between the study results. For smoking, significance was obtained only by the fixed-effects model (*p* = 0.002) and not by the random-effects model (*p* = 0.22), which typically takes into account the higher level of heterogeneity among different studies (Figure 5). However, the heterogeneity was always moderate, not high (*I*^2^ < 50%). The observed heterogeneity may stem from differences in study methodologies, such as definitions of smoking exposure, study populations, or follow-up duration.

In summary, the meta-analysis reveals the following:The evidence for HPV as a significant risk factor is robust, with the lack of heterogeneity further strengthening confidence in its oncogenic role.Smoking is confirmed as a significant risk factor, although the moderate degree of variability warrants careful consideration, especially given the limited number of studies.Alcohol consumption is not a relevant risk factor in this context.

### 3.5. Quality of Evidence

The GRADE approach highlighted that the evidence quality of our meta-analyses was moderate for smoking and low for HPV. All the studies included in our meta-analysis were observational and started with low confidence using the GRADE approach [14]. Only for smoking, the presence of a dose-response relationship, where increasing exposure or intervention leads to a proportional increase or decrease in the effect, resulted in upgrading the certainty of evidence by 1 level: from low to moderate [14,49]. For alcohol, the GRADE approach did not modify the initial low confidence level. In all meta-analyses, no downgrade was performed: there was low inconsistency (or heterogeneity) (*I*^2^ < 50%); no high-risk element was found for risk of bias (Table 1), indirectness of evidence, imprecision, or publication bias. Figure 7 illustrates how the GRADE approach was applied for smoking (inset 7A) and HPV (inset 7B).

## 4. Discussion

This study showed that smoking (*p* = 0.002) and HPV (*p* < 0.001) are risk factors for IP malignant transformation, with moderate and low quality of evidence, respectively. Alcohol as a risk factor did not reach statistical significance (*p* = 0.95), although cases examined for alcohol exposure were very few.

This study examined 26 articles with a total of 1271 inverted sinonasal papillomas, and 230 carcinomas arising in them. Collectively, the carcinoma incidence was 230/1271 (18.1%), which is considerably higher in comparison to that reported by other studies. However, the reported incidence of carcinoma in IP is extremely variable, from 4% [23] to 46.5% [48]. This is probably due to the presence of referral centers that preferably concentrate on more complex cases and do not reflect the actual incidence of the disease. Although these variations may have impacted the incidence of the disease in patients with IP, they should not have affected the evaluation of risk factors, which was the major subject of interest in our study.

The risk factors for IP malignant transformation have seldom been analyzed. Furthermore, only selected studies have precisely measured their impact. To measure the exact impact and to compare them, it was necessary to consider only studies with two different populations (exposed vs. non-exposed), both affected by inverted sinonasal papillomas, and to have the exact number of affected individuals in both categories and/or their odds ratio values. In the present study, only articles with all these precise measurements were considered to make comparison possible and to calculate different meta-analyses.

Smoking is a well-known risk factor in different oncogenic subsets, and it plays a primary oncogenic role also in head and neck carcinomas [48]. Despite its undoubted oncogenic role in IP incidence and recurrence, smoking has led to conflicting results. Deitmer [50] and Sham [5] found no significant difference between smokers and nonsmokers in terms of IP incidence, whereas Moon [6] reported that smoking was associated with IP recurrence after surgical excision. However, it is not surprising that smoking may be a risk factor for IP malignant transformation.

The present meta-analysis of smoking as a risk factor for IP malignant transformation showed moderate variability (*I*^2^ = 47%), warranting careful consideration and limiting its initial impact. The moderate variability was sustained by the fact that the significant role of smoking was present only using the fixed-effects model and not using the random-effects model, which typically takes into account more variability. Heterogeneity was low to moderate (*I*^2^ < 50%). However, the GRADE approach, which analyzes the quality of evidence, upgraded the smoking factor by 1 point, leading to a moderate quality of evidence, which further supports its role as a risk factor for IP malignant transformation. Therefore, smoking may be considered a risk factor for IP malignant transformation, and smoking cessation should be reasonably pursued in patients affected by IP.

HPV infection is also a well-known oncogenic risk factor in different anatomical settings: cervix, anus, and oropharynx [51]. Little is known about the role of HPV in the nasal cavity and paranasal sinuses. A recent study highlighted an increase in the incidence of HPV-associated sinonasal squamous cell carcinoma in sinonasal tract [52]. A meta-analysis showed that HPV may play a role in a specific subset of squamous cell carcinomas of the nose, especially in those subsites with high exposure to secretion flow (nasal cavity and ethmoid) [53]. The role of HPV infection in favoring IP development has been suggested, but this has not been fully proven [54]. However, HPV infection has been documented as an important risk factor for IP malignant transformation by three different meta-analyses [7,8,9]. The present study confirms the oncogenic role of HPV infection in IP malignant transformation, using both the fixed-effects model and the random-effects model. Vaccination (even if it is mainly limited to high-risk types) is reported to be useful in preventing oropharyngeal carcinoma [55]; therefore, it can be hypothesized that it may be useful in preventing malignant transformation in people affected by IP.

This is the first meta-analysis study that examines the role of risk factors other than HPV in determining IP malignant transformation. Previous meta-analyses have investigated only the role of HPV in IP malignant transformation; our results on the role of HPV are in line with data reported by three previously published meta-analyses [7,8,9]. Furthermore, the studies conducted by Ferreli et al. [9] and Zhao et al. [8] have also shown that high-risk HPV played a major role in IP malignant transformation. All previous meta-analyses conducted on this topic evaluated only the role of HPV in IP malignant transformation, whereas our study examined not only HPV but also other risk factors, such as smoking and alcohol.

Alcohol consumption did not reach statistical significance. However, only two studies were present in this meta-analysis; therefore, the small number of individuals may have affected this limited statistical role. In this view, even if there is definitely no evidence of a role for alcohol consumption as a risk factor in IP malignant transformation, there is also insufficient evidence to rule it out. Future studies may better clarify the role of alcohol and determine with more certainty whether it is a risk factor or not in IP malignant transformation. Alcohol plays a major role as an oncogenic driver in head and neck carcinomas [56], with the risk increasing with the intensity of drinking. In IP incidence, alcohol has seldom been analyzed. Sham reported a higher percentage of alcohol drinkers among the IP patients than among the controls (44% versus 33.3%), even if none of the odds ratios for the drinking risk factor in the subgroups studied were statistically significant [5].

Unfortunately, no data on the role of professional exposure in IP malignant transformation have been retrieved. Some studies have investigated the role of professional exposure in the incidence of IP [5,57], but not in IP malignant transformation. Data on professional exposure in this setting are very limited. All inclusion and exclusion criteria have probably limited the number of included studies. However, we did not want to consider biases and/or confounding factors, not to obtain information of limited value. In the meanwhile, age and sex were not considered, since they are unmodifiable factors. We preferred to focus on modifiable risks of malignancy, where preventive measures may be taken to limit the possibility of malignant transformation. Future studies, especially on professional exposure, are needed to clarify other important risk factors.

Limitations of this study are mainly related to the small number of studies conducted on this topic, which may reduce the validity of these meta-analyses. In addition, the low number of patients examined in some studies may have reduced the impact of the meta-analyses, especially in examining the role of alcohol. Statistical significance has considered the limited number of cases examined in each study, with a relative weight to the number of participants. Statistical analyses supported the collected data and may partially underline their validity.

## 5. Conclusions

In conclusion, our study demonstrated that smoking and HPV are risk factors for the malignant transformation of inverted sinonasal papillomas. Possible interventions include HPV vaccination and reducing or stopping tobacco smoking in people affected by sinonasal inverted papillomas. Further studies on risk factors, particularly on alcohol and professional exposure, are necessary to clarify this field.

## Figures and Tables

**Figure 1 cancers-17-01798-f001:**
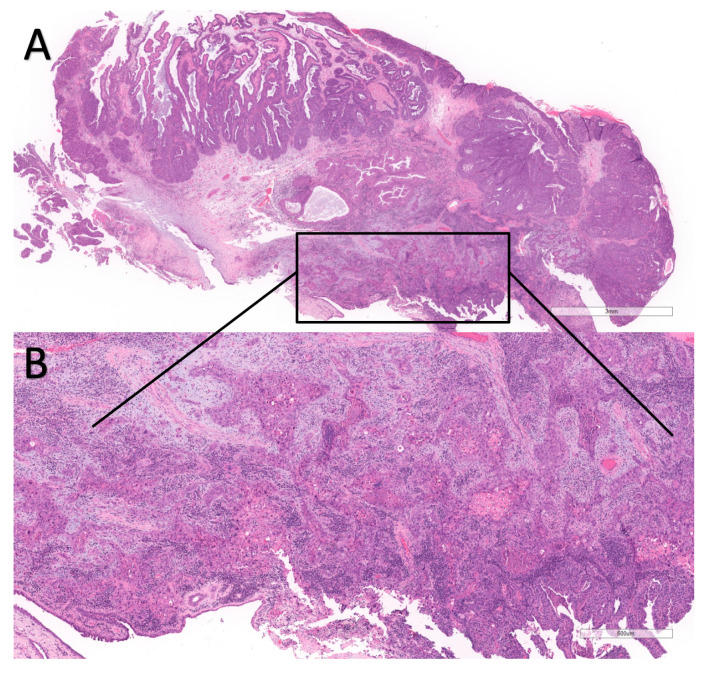
A case of squamous cell carcinoma arising in an inverted papilloma (IP). In the upper part (**A**), a low-power view is depicted. The lesion is polypoid and composed of an endophytic proliferation of epithelial cells, which can be diagnosed as an IP; however, at the base of the lesion, highlighted by the rectangle, a different lesion is observed. At high-power view (**B**), the highlighted area shows infiltrative margins, necrosis, keratinization, and atypical features, thus constituting a squamous cell carcinoma arising in an IP.

**Figure 2 cancers-17-01798-f002:**
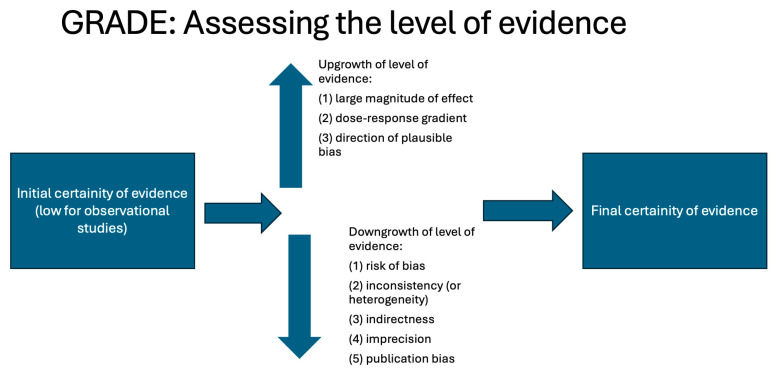
A summary of the GRADE (Grading of Recommendations, Assessment, Development, and Evaluation) approach for observational studies.

**Figure 3 cancers-17-01798-f003:**
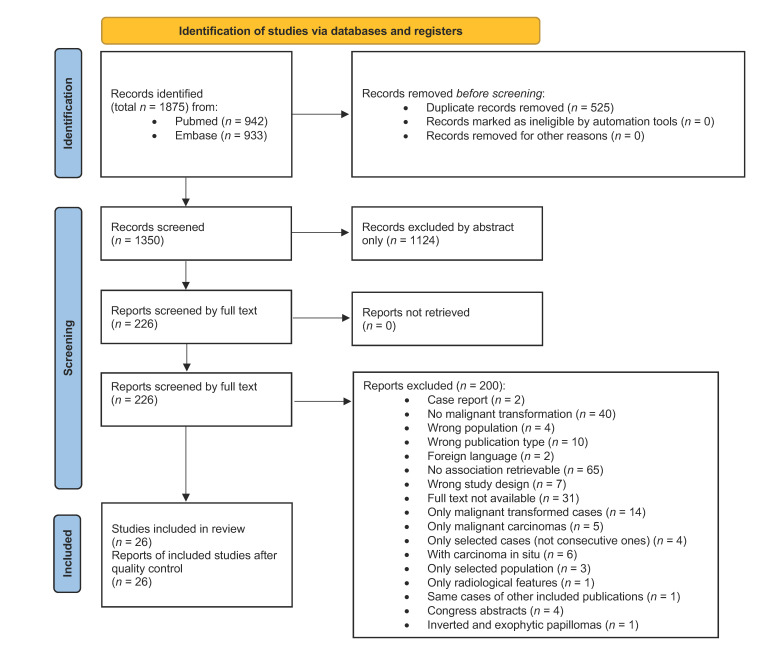
PRISMA flowchart highlighting the selection of articles.

**Figure 4 cancers-17-01798-f004:**
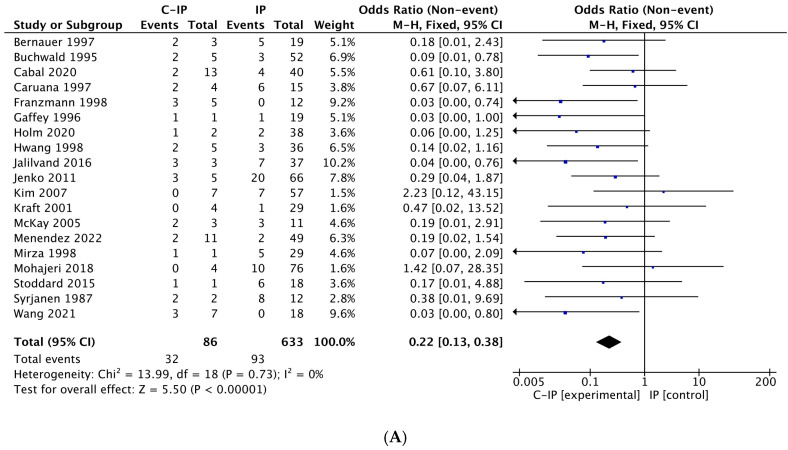
Forest plot and statistical data with fixed-effects model (**A**) and random-effects model (**B**) for HPV. Both models demonstrate that HPV is a risk factor for IP malignant transformation (*p* < 0.001). In both insets, the black diamond is located on the left side of the graph, supporting its role in malignant transformation (C-IP vs. IP). Legend: C-IP carcinoma in inverted papilloma, IP inverted papilloma [24,25,26,27,28,29,30,32,33,34,35,36,38,39,40,41,43,44,46].

**Figure 5 cancers-17-01798-f005:**
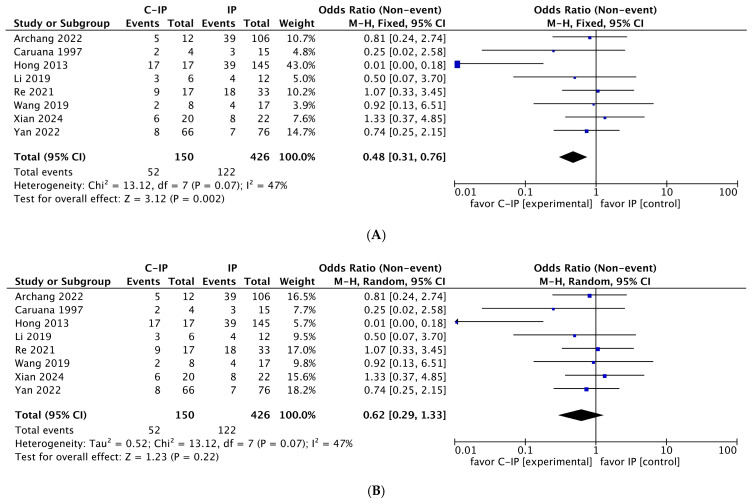
Forest plot and statistical data with fixed-effects model (**A**) and random-effects model (**B**) for smoking. The fixed-effects model (**A**) demonstrates that smoking is a risk factor for IP malignant transformation (*p* = 0.002). The random-effects model (**B**) did not confirm the significance of smoking (*p* = 0.22), probably because of moderate heterogeneity (*I*^2^ = 47%). In both insets, the black diamond is located on the left side of the graph, supporting the role of smoking in malignant transformation (C-IP versus IP). Legend: C-IP = carcinoma in inverted papilloma; IP = inverted papilloma [23,27,31,37,42,45,47,48].

**Figure 6 cancers-17-01798-f006:**
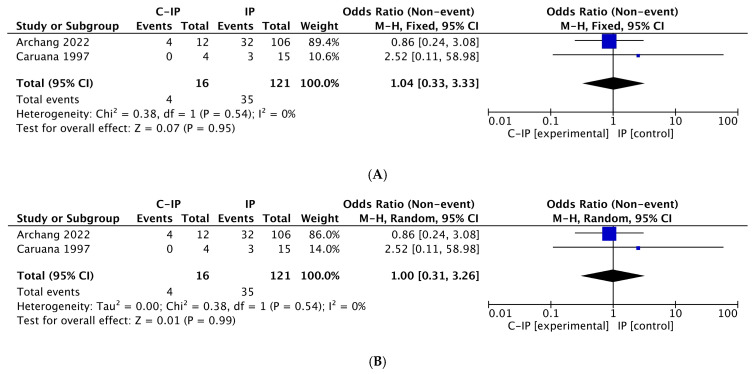
Forest plot and statistical data with fixed-effects model (**A**) and random-effects model (**B**) for alcohol. Both models demonstrate that alcohol is not a risk factor for IP malignant transformation. In both insets, the black diamond is located in the center of the graph, and the statistical test was not significant: *p* = 0.95 by fixed-effects model (**A**); *p* = 0.99 by random-effects models (**B**). Legend: C-IP = carcinoma in inverted papilloma; IP = inverted papilloma [23,27].

**Figure 7 cancers-17-01798-f007:**
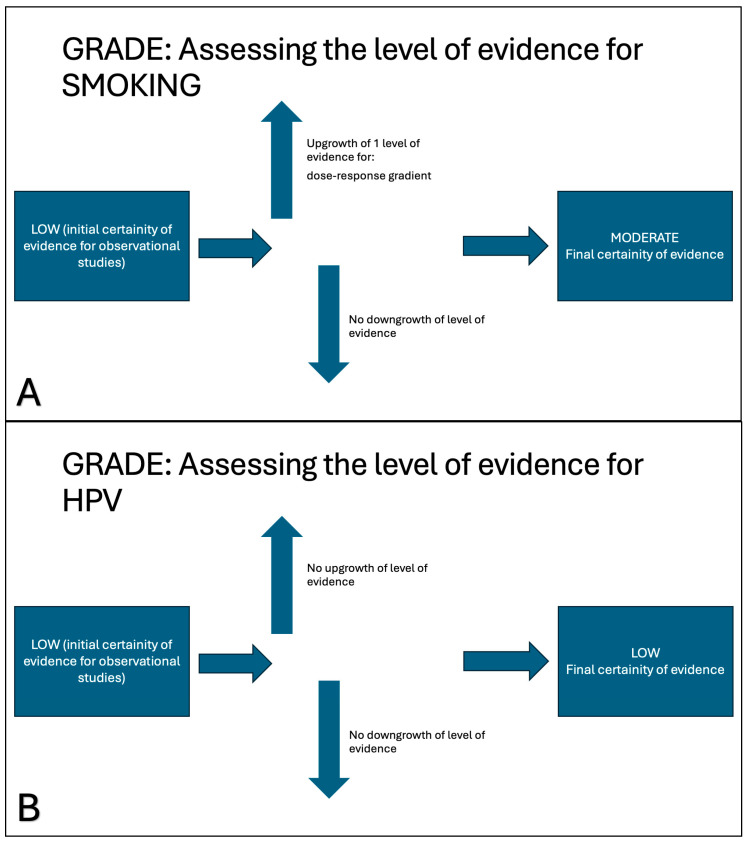
The GRADE approach was used to establish the quality of the evidence. In observational studies, the starting level was low. In smoking (**A**), the presence of a dose–response relationship increased the starting level by 1 point, with no downgrade, resulting in moderate-quality evidence. In HPV (**B**), neither upgrade nor downgrade was performed, resulting in low-quality evidence.

**Table 1 cancers-17-01798-t001:** The Newcastle–Ottawa Scale (NOS) was used to assess the risk of bias.

Author	1. Selection	2. Comparability	3. Exposure	Total Score
Archang [23]	4	1	3	8
Bernauer [24]	4	1	2	7
Buchwald [25]	4	1	3	8
Cabal [26]	4	1	2	7
Caruana [27]	4	1	2	7
Franzmann [28]	4	1	3	8
Gaffey [29]	4	1	2	7
Holm [30]	4	1	2	7
Hong [31]	4	1	3	8
Hwang [32]	4	1	2	7
Jalilvand [33]	4	1	2	7
Jenko [34]	4	1	2	7
Kim [35]	4	1	2	7
Kraft [36]	4	1	3	8
Li [37]	4	1	2	7
McKay [38]	4	1	2	7
Menendez [39]	4	1	2	7
Mirza [40]	4	1	2	7
Mohajeri [41]	4	1	3	8
Re [42]	4	1	2	7
Stoddard [43]	4	1	2	7
Syrjanen [44]	4	1	2	7
Wang [45]	4	1	3	8
Wang [46]	4	1	2	7
Xian [47]	4	1	2	7
Yan [48]	4	1	2	7
Average	4	1	2.3	7.3

**Table 2 cancers-17-01798-t002:** Cases with risk factors examined in the literature and with provided numbers of exposed vs. non-exposed individuals. The corresponding *p*-values (by odds ratio or Fisher’s exact test), as provided by the authors, are listed. Legend: IP = inverted papilloma; C-IP = carcinoma in inverted papilloma; N. = number; NA = not assessed; * depending on HPV types. Statistically significant values (*p* < 0.005) are highlighted in bold.

Authors	Year	N. of All IP	N. of IP Without Carcinoma	N. of C-IP	Rate of Malignancy (%)	HPV N. of Cases in IP vs. C-IP (*p*-Value)	Smoking Status N. of Cases in IP vs. C-IP (*p*-Value)	Alcohol N. of Cases in IP vs. C-IP (*p*-Value)	EBV N. of Cases in IP vs. C-IP (*p*-Value)
Archang [23]	2022	100	96	4	4/100 (4%)		39/106 vs. 5/12 (0.281)	32/106 vs. 4/12 (**0.035**)	
Bernauer [24]	1997	22	19	3	3/22 (13.6%)	5/19 vs. 2/3 (NA)			
Buchwald [25]	1995	57	52	5	5/57 (8.8%)	3/52 vs. 2/5 (NA)			
Cabal [26]	2020	55	41	14	14/55 (25.5%)	4/40 vs. 2/13 (NA)			
Caruana [27]	1997	19	15	4	4/19 (21.1%)	6/15 vs. 2/4 (NA)	3/15 vs. 2/4 (NA)	3/15 vs. 0/4 (NA)	
Franzmann [28]	1998	17	12	5	5/17 (29.4%)	0/12 vs. 3/5 (NA)			
Gaffey [29]	1996	20	19	1	1/20 (5%)	1/19 vs. 1/1 (NA)			1/12 vs. 1/1
Holm [30]	2020	53	50	3	3/53 (5.7%)	2/38 vs. 1/2 (NA)			
Hong [31]	2013	162	145	17	17/162 (10.5%)		39/145 vs. 17/17 (**<0.001**)		
Hwang [32]	1998	42	36	6	6/42 (14.3%)	3/36 vs. 2/5 (NA)			
Jalivand [33]	2016	40	37	3	3/40 (7.5%)	7/37 vs. 3/3 (**0.002**)			
Jenko [34]	2011	71	66	5	5/71 (7.0%)	20/66 vs. 3/5 (NA)			
Kim [35]	2007	57	50	7	7/57 (12.3%)	7/57 vs. 0/7 (NA)			
Kraft [36]	2001	34	30	4	4/34 (11.8%)	1/29 vs. 0/4 (NA)			
Li [37]	2019	25	16	9	9/25 (36.0%)		4/12 vs. 3/6 (NA)		
McKay [38]	2005	14	11	3	3/14 (21.4%)	3/11 vs. 2/3 (NA)			
Menendez [39]	2022	60	49	11	11/60 (18.3%)	2/49 vs. 2/11 (**0.034 0.006** *)			
Mirza [40]	1998	30	29	1	1/30 (3.3%)	5/29 vs. 1/1 (NA)			
Mohajeri [41]	2018	76	72	4	4/76 (5.3%)	10/76 VS. 0/4 (NA)			
Re [42]	2021	50	33	17	17/50 (34.0%)		18/33 VS. 9/17 (NA)		
Stoddard [43]	2015	19	18	1	1/19 (5.3%)	6/18 VS. 1/1 (≥5 mRNA) (NA)			
Syrjanen [44]	1987	14	12	2	2/14 (14.3%)	8/12 vs. 2/2 (NA)			
Wang [45]	2019	25	17	8	8/25 (32.0%)		4/17 vs. 2/8 (NA)		
Wang [46]	2021	25	18	7	7/25 (28.0%)	0/18 vs. 3/7 (NA)			
Xian [47]	2024	42	22	20	20/42 (47.6%)		8/22 vs. 6/20 (0.662)		
Yan [48]	2019	142	76	66	66/142 (46.5%)		7/76 vs. 8/66 (0.596)		
**Total**		1271	1041	230	230/1271 (18.1%)				

## Data Availability

No new data were created. A summary of all meta-analyses is provided in tables and figures.

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
