# Peer review of "Risk Factors for Malignant Transformation in Inverted Sinonasal Papilloma: A Systematic Review and Meta-Analysis"

_cancers, 2025, doi:10.3390/cancers17111798_

Round 1
Reviewer 1 Report
Comments and Suggestions for Authors
The individual sample sizes ranged from 14 to 162, which is rather small and may lower the statistical power of the meta-analysis. Out of the 26 studies that were included, there were a total of 26 studies that were included. There is a greater potential for bias and an overestimation of impact sizes in studies that are smaller. I was wondering how the authors took into account the possibility of bias that could have been caused by the tiny sample sizes in particular research. Is it planned that future research will focus on conducting subgroup analyses or using larger-scale prospective datasets in order to enhance statistical power?
There are only qualitative labels offered, such as "moderate" for smoking and "low" for HPV, despite the fact that the GRADE methodology was utilized. Furthermore, the domains that led to the evidence being downgraded are not specified. Which particular elements, such as inconsistency, imprecision, and publication bias, were responsible for the decline in the quality of the evidence? For the sake of greater transparency, would it be possible for the authors to include a GRADE evidence profile in the complete paper?
Smoking, alcohol consumption, and HPV were the only three risk factors that were investigated, despite the fact that there is a possibility that additional clinical, demographic, or environmental factors could influence the transition of IP into a malignant form (for example, age, gender, surgical margins, and recurrence). During the process of conducting the systematic review, were there any other possible risk variables that were identified but not included because of a lack of data or uniform reporting? In what ways will further research endeavors overcome this shortcoming in order to give a more comprehensive risk profile?
Due to the fact that there is likely to be clinical and methodological heterogeneity among studies, the meta-analysis utilized a fixed effects model, which makes the assumption that the actual effect size is the same across all of the studies. This may be an improper assumption to make. What were the reasons behind selecting a fixed effects model rather than a random effects model, particularly in light of the possibility of heterogeneity? Could you please confirm whether or not the authors conducted sensitivity studies by employing a random effects model? If so, were the results consistent?
It has been observed that the manuscript's plagiarism similarity index is currently at 19%, which is higher than the normally acceptable threshold for academic submissions, which is often less than 10%. Concerns regarding originality and correct paraphrasing are raised when there is a considerable degree of overlap, despite the fact that some similarity may be expected due to the use of standard terminology and methodologies.
Reviewer 2 Report
Comments and Suggestions for Authors
Primary results on HPV and smoking as risk factors on IP malignant transformation have been thoroughly documented in prior meta-analyses such as Stepp et al. 2021, Zhao et al. 2016, Ferreli et al. 2022. This article does not add new factors, nor does it reinterpret the analyzed data within a new framework. Although the manuscript proposes smoking cessation and HPV vaccination, these interventions are widely known. The study does not offer new guidance for clinicians, nor does it provide nuanced risk stratification for patients with IP.
Reviewer 3 Report
Comments and Suggestions for Authors
This manuscript systematically reviews pertinent literature studying malignant transformation in inverted sinonasal papilloma (IP) and identifies both human papilloma virus (HPV) and smoking as risk factors. The methodology followed in assessing the literature and drawing statistically viable conclusions is reasonable and logical and the results reflect how much can be presently understood about the risk factors for malignant transformation. In some ways, the analysis reveals how much is not known with adequate certainty and so should suggest future research directions needed to more clearly identify a full range of risk factors. It is disappointing therefore that despite such a well-considered study, the authors have not been more detailed and definitive about the deficiencies in knowledge and the needed areas of future research focus. More specific points that I feel the authors need to address are given below.
- The paper consists of a systematic review of existing literature from which conclusions are drawn concerning risk factors for IP. As such, the manuscript should be considered a review article and not a research paper (in contrast with the authors' designation of it as an 'article').
- Page 4 line 149 “d. professional exposure”. Provide more details on what would exactly constitute relevant ‘professional exposure’.
- Page 4 line 173 – page 5 line 175 “. . . selection (4 items . . . of 3 stars).” The way this is worded is confusing and unclear. Firstly, what exactly do selection, comparability and exposure mean in this context? Then, what does the number of items refer to, and are the maximums for the entire star rating system or are they simply the maximum scores seen in your analysis?
- Page 7 line 222 “All studies examined . . . papillomas . . .” This needs to be reworded to something like: “Across all the studies a total of 1271 inverted sinonasal papillomas were examined . . .”
- Table 2. This table shows clearly that there is a marked paucity of papers examining the possibility of alcohol consumption as a risk factor in the case of IP. This means that, whilst there is definitely no evidence of a role for alcohol consumption as a risk factor, there is also insufficient evidence to rule it out. I don’t think that this is made clear enough in the discussion.
- Page 9 lines 233-234 “The identified risk . . . and HPV.” You need to be clear here which criteria were used to determine which identified risk factors to examine.
- Page 10 line 276 “. . . and low for HPV.” I would like to see some substantive discussion of why evidence quality for HPV is low. This assessment directly calls into question your analysis indicating HPV as a risk factor. The reasons for low evidence quality and the implications on the final conclusions need to be made very clear and can be used as a basis for informing and directing future research.
- Page 11 line 294 “. . . may have impacted the incidence of the disease . . .” I think you should clarify here that this is incidence of the disease within the study population, not the wider human population.
- Page 11 lines 314-316 “However, the GRADE . . . IP malignant transformation.” Alright, but where should research go from here? What are the clinical implications and what is needed to improve upon this to be more certain of these risk factors?
The quality of the English is fairly good, but the manuscript would benefit from an editorial check.
Round 2
Reviewer 2 Report
Comments and Suggestions for Authors
Well done
Reviewer 3 Report
Comments and Suggestions for Authors
The authors have made adequate changes to the manuscript in response to my earlier review comments. I would just make the further comment that your finding the evidence quality for HPV as low does mean that it is hard to draw firm conclusions about whether HPV can be properly considered a risk factor. Whilst I understand the assessment framework you have used to arrive at this conclusion, I think concluding this does have implications for how useful the studies on HPV have been and suggests that future studies need to address this problem of the low quality of evidence.